# Evaluation of Some Composite Paint Coatings' Appearance Quality Using Fractal Dimension

Valentina Loganina [1], Roman Fediuk [2,3,*], Valery Lesovik [3,4], Mugahed Amran [5,6,*], Diyar N. Qader [7], Olga Litvinets [2], Viktoria Okhotkina [2], Raizal S. M. Rashid [8], Maksim Lomov [3] and Irina Moskovaya [2]

[1] Department of Quality Management and Technology of Construction Production, Penza State University of Architecture and Construction, 440028 Penza, Russia

[2] Department of Geoinformation Technologies, Polytechnic Institute, Far Eastern Federal University, 690922 Vladivostok, Russia

[3] Department of Building Materials Science, Products and Structures, Belgorod State Technological University Named after V.G. Shoukhov, 308012 Belgorod, Russia

[4] Central Research and Design Institute of the Ministry of Construction, Housing and Utilities of the Russian Federation, 119331 Moscow, Russia

[5] Department of Civil Engineering, College of Engineering, Prince Sattam Bin Abdulaziz University, Alkharj 16273, Saudi Arabia

[6] Department of Civil Engineering, Faculty of Engineering and IT, Amran University, Amran 9677, Yemen

[7] Department of Civil Engineering, Cihan University-Erbil, Erbil 44001, Kurdistan Region, Iraq

[8] Department of Civil Engineering, Faculty of Engineering, Universiti Putra Malaysia, Serdang 43400, Selangor, Malaysia

* Correspondence: roman44@yandex.ru (R.F.); m.amran@psau.edu.sa (M.A.)

**Abstract:** Composite materials are characterized by multiple layers, which leads to a complexity in the design in order to ensure the effective operation of the constituent elements. This article provides information on the use of fractal dimension in assessing the quality of the appearance of paint coatings. The scientific originality of the article lies in the establishment of a correlation between the surface roughness of coatings, the quality grade of their appearance and fractal dimension. As a result, a model of the length of the coating surface profile, with the fractal dimension *D*, was proposed. The practical significance lies in the proposal to evaluate the quality of the surface of paint and varnish coatings in terms of fractal dimension. An increase in the surface roughness of the coating, a decrease in the appearance quality grade and an increase in the fractal dimension have been observed. Numerical values of the index of the fractal dimension of the coating surface profile, which depended on the porosity of the substrate, have been obtained. The influence of the filling of the paint composition on the quality of the appearance of the coatings has been estimated. It has been revealed that there was an increase in the surface tension of the paint composition, a decrease in the quality of the appearance of the resulting coating and an increase in the roughness and fractal dimension of the coating surface. The influence of the method of applying the paint composition and the preparation of the base surface on the quality of the appearance of the coatings are considered. The results obtained can be applied in various types of production to improve the quality of paint coatings.

**Keywords:** composites; coatings; appearance; quality; regularities; formation; model

## 1. Introduction

To increase the architectural expressiveness and performance of the external concrete walls of buildings, various types of finishes are used [1]. One of the most common types of finishing is painting, its share among all types of finishing materials is more than 50% [2]. Coatings for facade finishing, performing aesthetic-protective functions, must have a high-quality appearance [3]. The quality of the appearance means the absence of defects (blotches, smudges, shagreen, strokes and scratches, waviness and variability) on the paintwork [4].

The presence of certain defects on the surface of the paintwork characterizes the quality class of the appearance [5].

The definition and evaluation of the quality of paint coatings is a subjective process. To date, there have been various methods used for this. Vaspero et al. [6] applied the electron microscopy method to the quality control of paint coating surfaces. Zhong et al. [7] used the multi-modal plasma-focused ion beam serial section tomography of an organic paint coating. Oliveira and Ferreira ranked high-quality paint systems that were either intact [8] or defective [9], using electrochemical impedance spectroscopy. Quevauviller [10] summarized certified reference materials for the quality control of inorganic analyses of manufactured paint coatings. Zhou et al. [11] developed a mathematical model for characterizing the full process of volatile organic compound emissions from paint film being coated on porous substrates. Aktas et al. [12] systematically investigated coating application methods and soft paint types to detect cavitation erosion on marine propellers. Tanzim et al. [13] generated a prediction model of fan width by optimizing the paint application process for electrostatic rotary bell atomizers.

These differences in methods force us to be guided by standards for assessing the quality of paint coatings. In accordance with the Russian standard GOST 9.032-74 [14], seven quality classes of the appearance of coatings on a metal substrate are defined:

I grade—no defects can be allowed for high-gloss, glossy, semi-glossy and semi-matt. For matt coatings, no more than 4 inclusions per m$^2$;

II–VII grades—possible weed or individual inclusions, taking into account their number (pcs/m$^2$) and depending on the length, width and diameter of the defect and the distance between them (mm), as well as notching.

In addition to the above defects, the quality III–VII grades admit to waviness, V–VII—streaks, IV–VII—off-shade.

In works [15,16], the quality of the appearance of the protective-decorative coatings for cement concrete are evaluate IV–VII grades (Table 1).

**Table 1.** Permissible defects for coatings depending on their grades.

| Coating Grade | Defect Name | Regulatory Requirements for Coatings | | |
|---|---|---|---|---|
| | | Gloss | Semi-Matt | Matt |
| IV | inclusions: number of pcs/dm$^2$ | 1 | 1 | 1 |
| | size, mm | ≤1.0 | ≤1.0 | ≤1.0 |
| | distance between inclusions, mm | ≤10 | ≤10 | ≤10 |
| | shagreen | allowed | allowed | allowed |
| | streaks | not allowed | not allowed | not allowed |
| | strokes, notching | allowed | allowed | allowed |
| | waviness, mm | ≤2 | ≤2 | ≤2 |
| | off-shade | not allowed | not allowed | not allowed |
| V | inclusions: number of pcs/dm$^2$ | 4 | 4 | 5 |
| | size, mm | 2.0 | 2.0 | 2.0 |
| | shagreen | allowed | allowed | allowed |
| | streaks | not allowed | not allowed | not allowed |
| | strokes, notching | allowed | allowed | allowed |
| | waviness, mm | ≤2.5 | ≤2.5 | ≤2.5 |
| | offshade | not allowed | not allowed | not allowed |
| VI | inclusions: number of pcs/dm$^2$ | 8 | 8 | 8 |
| | size, mm | 3.0 | 3.0 | 3.0 |
| | shagreen | not allowed | not allowed | not allowed |
| | streaks | allowed | allowed | allowed |
| | strokes, notching | allowed | allowed | allowed |
| | waviness, mm | ≤4.0 | ≤4.0 | ≤4.0 |
| | off-shade | allowed | allowed | allowed |

It is necessary to search for new, simpler, methods of quality control of painted coatings. The quality of the surface of the paintwork can be assessed, including its roughness along the surface profile. It seems promising to use the methods of fractal physics, which are currently widely used in evaluating the structure of polymers, to assess the surface profile of the coating [17–22].

The successful application of fractal analysis methods to the study of polymers means that there is hope that this approach can be successfully extended to assess the quality of the appearance of coatings [23–27].

Fuseini and Zaghloul [28–30] completed statistical and qualitative analyses of the kinetic models using electrophoretic deposition of polyaniline. Researchers [31–33] developed polyester composite materials using natural fibres and nano fillers. Takada et al. [34] used fractal dimensional analysis on the dispersion/aggregation state of multi-walled carbon nanotubes (MWCNT) in polystyrene. For example, the effect of UV-induced polymer-MWCNT chemical bond formation and its influence on electrical conductivity of their composites was researched. Zivic et al. [35] investigated the self-interacting polymer chains terminally anchored to adsorbing surfaces of three-dimensional fractal lattices. Li et al. [36] developed a fractal-crazing constitutive model of glassy polymers which considered damage and toughening. Marquardt et al. [37] researched the fractal dimensions of blast-cleaned steel surfaces and their effects on the adhesion of polymeric foil systems with integrated pressure-sensitive adhesives. Pandey and Chandra [38] investigated the effect of thermal degradation of polymer on redox-initiated fractal geometries. The mechanics of the studied materials and their mechanical characteristics are of great importance in this sense [39].

Despite the good results obtained as a result of the use of fractal dimension methods for modeling various polymers, studies of similar methods on paint coatings are not enough, which is a research problem. For the coating surface profile, the fractal dimension $D$ is in the range $1 < D < 2$ [40–43]. The greater the roughness of the coating, the more curved the coating profile and the greater the $D$ value. Thus, the fractal dimension $D$ of the coating profile can serve as a criterion for the quality of its appearance, reflecting the presence of inclusions, stripes and waviness.

In this article, an attempt was made to evaluate the possibility of describing the quality of the appearance of coatings using fractal dimension. To achieve this goal, the following tasks were solved: establishing a correlation between the surface roughness of the coating, the appearance quality class and the fractal dimension; obtaining the numerical values of the index of the fractal dimension of the coating surface profile, which depend on the porosity of the substrate; assessing the effect of filling the paint composition on the quality of the appearance of coatings, and considering the influence of the method of applying the paint composition and the preparation of the base surface on the quality of the appearance of the coatings.

## 2. Materials and Methods

### 2.1. Characteristics of the Raw Materials Used

The choice of the paints used in research was due to their wide prevalence in the finishing of building products and structures. At the same time, the selected paints had a different base, which allowed for a wider coverage in terms of the practical applicability of the research results. The following paint compositions were used in the work: alkyd enamel PF-115 (Yaroslavl colors, Yaroslavl, Russia); oil paint MA-15 (Lakra, Moscow, Russia), nitrocellulose paint NC-132 (Teks, Saint-Petersburg, Russia); acrylate paint Universal (Teks, Saint-Petersburg, Russia); acrylic water-dispersion paint Facade (Belyi dom, Kaluga, Russia); polyvinyl acetate cement (PVAC), polymer–lime; cement–perchlorovinyl (CPCV), and lime paint. For the preparation of colorful compositions, the following materials were used:

— White Portland cement (Belgorod cement, Belgorod, Russia), factory-made colored cements (Yaroslavsky pigment, Yaroslavl, Russia);
— Quartz sand (Volsk, Russia) for decorative powder;

- — Lime–fluff (Penza, Russia);
- — Lime paste, 50% (Penza, Russia);
- — Polyvinyl acetate dispersions (PVAD), grades DF 47/70, and 48/50 (Penza, Russia);
- — Liquid glass (Penza, Russia);
- — Water-repellent liquid 136-41 (Penza, Russia).

Expanded clay–concrete M75, cement–sand mortar, heavyweight concrete, expanded clay–concrete and glass were used as substrates.

### 2.2. Laboratory Equipment and Research Methods

Dynamic viscosity of the paint was determined by the Stokes method. To do this, the test liquid was poured into a graduated cylinder, a metal ball of known density $\rho_{ball}$ and radius $r$ was brought to the surface and lowered (without a push). As soon as the movement of the ball became uniform, the electric second timer was turned on and the lowered time $t$ of the ball was determined. The viscosity of the liquid was given by the formula:

$$\mu = \frac{2}{9} g r^2 \frac{(\rho_{ball} - \rho_{paint)t}}{l} \tag{1}$$

where:

$g$—acceleration of gravity m/s$^2$;

$t$—time during which the ball passes the distance between the marks A and B, s;

$l$—distance between the marks $A$ and $B$, cm;

$\rho_{ball}$, $\rho_{paint}$—density of the ball and paint, g/cm$^3$, respectively.

The conditional viscosity of the paint compositions was determined using a VZ-1 viscometer (K-M, Saint-Petersburg, Russia). The method determined the duration of the expiration (in seconds) of a certain volume of paint material through a nozzle of a given size. The conditional viscosity was taken by measuring the duration of the outflow of 100 mL of material through a nozzle with a diameter of 4 mm.

The determination of surface tension was carried out by using the method of counting drops (the stalagmometric method).

The determination of the flowability (pouring) of colorful materials was carried out according to the following method: applying five pairs of parallel strips of material and determining the degree of spreading by the resulting number of merged strips. The strips were applied with a special device. The degree of spreading of the five pairs of parallel strips was assessed on a ten-point "pouring" scale.

The gloss of the coatings was determined by the photoelectric method on an FB-2 gloss meter (K-M, Saint-Petersburg, Russia), according to the Russian standard GOST 896-69.

The coating profile (roughness) was determined using an A1 type profilometer-profilograph (K-M, Saint-Petersburg, Russia), according to the Russian Standard GOST 19300-86, with measurement limits of 0.00002–0.25 mm. The length of the coating surface profile was determined using a curvimeter. The fractal dimension of the surface profile of the coatings was estimated using a geometric method. To do this, the image of the curve was obtained using a profiler-profilometer covered with a grid consisting of squares with a side L1. Then, the number of squares N, through which the curve N (L1) passed, was counted. By changing the scale of the grid each time the number of squares intersected, the curve N (L2), N (L3) ... N (Ln) was counted again. Then, the dependence N (L) was plotted in double logarithmic coordinates, the slope of which was used to determine the fractal dimension. The quality of the substrates was evaluated by the index of total porosity P and surface porosity $P_s$. Surface porosity was determined by the ratio of the sum of the pore area to the total area of the measured surface. The pore diameter was measured using a measuring magnifier ×24.

To substantiate the conclusions about the correlation between surface roughness, quality class and fractal dimension, a statistical analysis was carried out. A total of 50 measurements were taken on each surface (Figure 1). To assess the uniformity of the distribution of roughness indicators along the strike, statistical indicators were calculated.

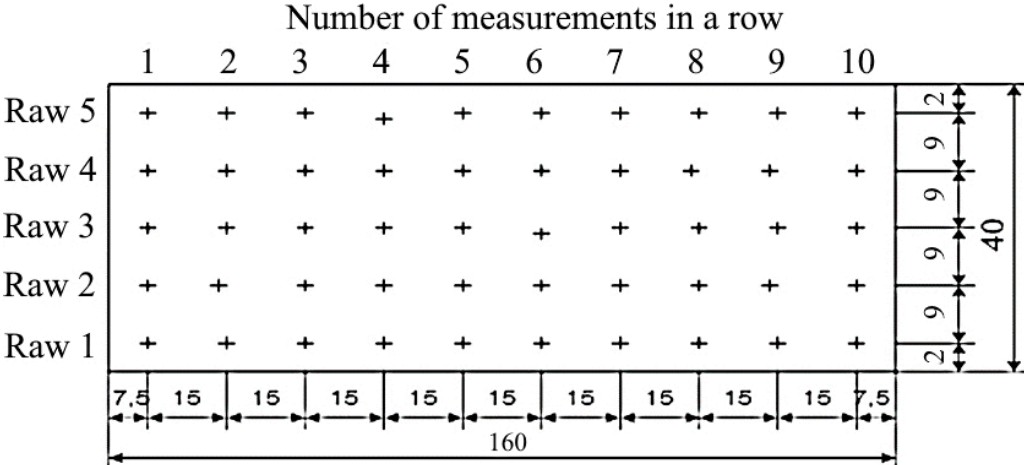

**Figure 1.** Scheme for measuring the surface roughness of paint coatings on a cement substrate (all dimensions are given in mm).

### 3. Results and Discussion

Table 2 shows the results of assessing the quality of the appearance of the coatings, carried out in accordance with the Russian standard GOST 9.407-2015 and having used the fractal dimension of the surface profile of the coatings.

**Table 2.** Assessment of the quality of the appearance of the coating.

| Paint | Coating Surface Roughness, $R_a$, μm/Coefficient of Variation, % | Fractal Dimension, *D* | Surface Profile Perimeter, mm | Coating Surface Quality Grade | Gloss, % |
|---|---|---|---|---|---|
| PF-115 | 0.22/3.0 | 1.17 | 106 | V | 70.8 |
| | 0.75/4.0 | 1.35 | 116 | VI | 67.7 |
| MA-15 | 0.2/2.0 | 1.075 | 102 | V | 83.1 |
| | 0.48/3.0 | 1.125 | 109 | VI | 78 |
| NC-123 | 0.2/2.1 | 1.1 | 105 | V | 76.9 |
| | 0.75/3.9 | 1.36 | 121 | VI | 67.7 |
| Universal | 0.75/4.0 | 1.3 | 216 | VI | 66.2 |
| | 1.26/4.2 | 1.42 | 221 | VII | 64.2 |
| Facade | 0.6/3.4 | 1.25 | 192 | VI | 66.2 |
| | 1.24/4.1 | 1.7 | 225 | VII | 62.2 |

As is displayed, there was a correlation between the coating surface roughness, the appearance quality grade and fractal dimension. As the surface roughness increased, the appearance quality grade decreased and the fractal dimension *D* increased. For example, the coating surface profile fractal dimension based on PF-115 had a coating surface roughness and appearance quality class of 0.2 μm and V was *D* = 1.17, respectively. While the coating surface roughness and the appearance quality class was 0.75 μm and VI was *D* = 1.35, respectively. These similarities are typical for other types of coatings. The change in gloss and the coating surface profile parameters on the sample length (equal to 10 cm) also proved this fact. When the coating surface profile fractal dimension increased, the gloss decreased and the numerical values of the profile perimeter increased.

The correlation between the length *L* of the coating surface profile on the sample length *l* and the fractal dimension *D* can be approximated as

$$L = lD^{bD} \tag{2}$$

where *b* is a constant factor. Formula (2) was obtained by the least squares method.

In particular, for the coatings based on paint PF-115 the expression (2) is

$$L = 99.62D^{0.33D}$$

—  For paint MA-15-based coating

$$L = 99.08D^{0.63D}$$

—  For paint NC-based coatings

$$L = 99.55D^{0.416D}$$

—  For acrylic paint-based coating

$$L = 115.28D^{1.47D}$$

—  For water-dispersible façade paint-based coatings

$$L = 131D^{0.645D}$$

The correlation coefficient was 0.93–0.95.

The results of the research suggest that, for fractal dimension $D$ = 1–1.09, the coating appearance quality grade was IV, for D = 1.1–1.2 is V, for D = 1.21–1.4 is VI, for D = 1.41–1.99 is VII. By applying the integral performance index to their appearance, fractal dimension let us estimate the paint coatings' quality more objectively. The results of calculations in accordance with model (2) were in good agreement with the data given in Table 2.

Research has shown that the quality of the appearance of the coatings was determined by the nature of the bottling of the paint. Bottling is considered as a rheological process, which can be described by the following expression:

$$h = \frac{b^2}{8\sigma}f \tag{3}$$

where
   $h$—stroke height,
   $b$—width of stroke,
   $f$—shift limit stress of the paint,
   $\sigma$—surface tension of the paint,
   Formula (3) is presented in [44].

Analysis of the data from Table 3, indicates that there was some slowing down of the time of restoration of the structure of paint when applied to the porous surface of the mortar. For example, on a glass substrate, the time for restoring the structure of the PF-115 paint composition was 3 min, and on the cement substrate it was 5 min, while filling in both cases was satisfactory. Undoubtedly, the time to restore the structure of the paint composition depended both on the porosity of the substrate and on the rheological properties of the paint.

The evaluation results of the coatings' surface profile indicate that with an increase in the surface tension of the paint composition, a lower quality in the appearance of the coating formed was observed. Thus, with the surface tension of an acrylate paint Universal, equal to s = 36.13 mJ/m$^2$, the surface roughness of the coating was $R_a$ = 2.4 μm, and the fractal dimension was $D$ = 1.53. Similar patterns were also observed in other types of coatings.

**Table 3.** Effect of rheological properties of paints on the coatings appearance quality.

| Paint | Surface Roughness *, $R_a$, μm/Coefficient of Variation, % | Fractal Dimension of the Coating Surface, $D$ | Surface Tension of the Paint Composition, mJ/m$^2$ | Dynamic Viscosity of the Paint Composition, Pa·s | Filling the Colorful Composition **, min. |
|---|---|---|---|---|---|
| Alkyd paint (PF-115) | 0.58/3.2 | 1.29 | 19.35 | 7.92 | ≤5/≤5 |
| | 0.4/3.1 | 1.18 | 18.37 | 6.86 | ≤5/≤3 |
| | 0.21/2.2 | 1.06 | 16.67 | 5.8 | ≤5/≤3 |
| Oil paint (MA-15) | 0.8/4.3 | 1.3 | 17.48 | 23.68 | ≤5/≤3 |
| | 0.69/3.7 | 1.1 | 16.93 | 14.8 | ≤5/≤3 |
| | 0.46/3.3 | 1.03 | 16.18 | 10.36 | ≤5/≤3 |
| Nitrocellulose (NC-132) | 0.78/4.0 | 1.3 | 27.09 | 14.02 | ≤5/≤3 |
| | 0.6/3.3 | 1.12 | 24.08 | 7.38 | ≤5/≤3 |
| | 0.32/2.7 | 1.06 | 22.12 | 6.39 | ≤5/≤3 |
| Acryl water dispersed (Façade) | 3.01/2.6 | 1.4 | 37.37 | 40.04 | ≤15/≤15 |
| | 2.55/2.4 | 1.25 | 34.96 | 30.8 | ≤15/≤15 |
| | 1.85/2.0 | 1.1 | 31.88 | 21.56 | ≤15/≤15 |
| Acrylate (Universal) | 2.4/2.1 | 1.53 | 36.13 | 33.44 | ≤15/≤15 |
| | 1.77/2.0 | 1.3 | 33.87 | 24.32 | ≤15/≤15 |
| | 1.44/2.0 | 1.1 | 30.96 | 15.2 | ≤15/≤15 |

* coating roughness was evaluated on a glass substrate; ** on the left are the numerical values of the recovery time of the paint structure when applied onto a substrate from the mortar, on the right are onto a glass substrate.

Further studies were aimed at assessing the probability of coating failure due to the presence of surface defects. Suppose there is one defect (inclusions, stripes, etc.) on the surface of the coating, which has a random location. Let us divide the area of the coating under study into n sections, the area of which is equal to the area of the defect. The probability of a defect in a certain area during operation is equal to P. In this case, the condition must be observed that in real time $np = const$. A coverage is considered "failed" if more than $n_{def}$ areas are found on its surface, which is determined from the expression:

$$n_{def} = \frac{S_{def}n}{100} \tag{4}$$

$$F(c) = \sum_{x=0}^{S_{def}n} \frac{c^x}{x!}e^{-c} \tag{5}$$

where $c = np = const$.

An assessment was made of the kinetics of the concentration of defects on the surface of coatings during their aging and the probability of destruction was estimated in relation to the quality of their appearance, characterized by fractal dimension and roughness. After curing the coatings, the colored solution samples were subjected to alternate freeze–thaw. The different quality of the appearance of the coatings was created by changing the porosity of the substrate and the rheological properties of the paint compositions. The defect concentration was determined on a coating area of 64 cm$^2$.

It was found that cracks appeared locally and formed near defects on the coating surface. In particular, on the coating based on paint MA-15, characterized by a fractal dimension and roughness index of 1.089 and $R_a = 0.23$ μm, respectively, surface cracks appeared, visible to the naked eye, after 5 freeze–thaw cycles, and on a coating with an index of fractal dimension and roughness 1.069 and $R_a = 0.14$ μm, respectively, after 15 test cycles. These patterns were characteristic of other coatings (Table 4).

Summarizing the above, it can be stated that, by adjusting the rheological and techno-logical paint composition, the physical and mechanical properties of the substrate, etc., it is possible to obtain coatings characterized by a high-quality appearance grade.

**Table 4.** Change in the quality of the appearance of coatings in the process of freezing–thawing.

| Paint Coating | Surface Roughness, $R_a$, μm/Fractal Dimension | Number of Defects after Test Cycles/Destruction Probability, % | | | | |
|---|---|---|---|---|---|---|
| | | 0 | 5 | 10 | 13 | 15 |
| Alkyd paint (PF-115) | 0.12/1.109 | 36/16.8 | 39/42.7 | 50/100 Coating peeling off | - | - |
| | 0.10/1.1 | 30/15.1 | 32/38.3 | 36/42.1 | 68/100 Coating peeling off | - |
| | 0.08/1.08 | 18/13.7 | 25/35.5 | 31/40.7 | 56/47.2 | 57/53.8 |
| | 0.21/1.17 | 45/17.0 | 62/100 Coating peeling off | - | - | - |
| | 0.18/1.12 | 30/15.9 | 47/100 Coating peeling off | - | - | - |
| | 0.15/1.11 | 22/14.1 | 46/30.9 | 54/41.2 | 59/45.4 | 63/54.3 |
| | 0.26/1.18 | 54/17.6 | 67/33.4 | 80/100 Coating peeling off | - | - |
| Oil paint (MA-15) | 0.23/1.089 | 29/25.0 | 60/100 Coating peeling off | - | - | - |
| | 0.18/1.073 | 20/24.3 | 23/35.1 | 26/38.1 | 30/40.0 | 33/41.3 |
| | 0.14/1.069 | 10/23.2 | 15/34.0 | 20/35.2 | 21/39.2 | 26/40.8 |
| | 0.26/1.2 | 30/25.2 | 72/100 Coating peeling off | - | - | - |
| | 0.20/1.075 | 23/24.5 | 49/100 Coating peeling off | - | - | - |
| | 0.17/1.073 | 12/23.6 | 16/34.4 | 21/36.3 | 25/39.6 | 29/40.2 |
| | 0.30/1.24 | 34/25.6 | 82/100 Coating peeling off | - | - | - |

The quality of the appearance of coatings is influenced by the porosity of the substrate (Table 5). Analysis of the data given in Table 4 shows that, with an increase in the porosity of the substrate, the index of the surface roughness of the coatings increased. For example, the surface roughness of lime coatings increased to 83.4 μm (the porosity of the substrate was 9.6%), for polymer lime it went up to 88.1 μm (substrate porosity 11.9%) and for PVAC coatings it went up to 46.2 μm (substrate porosity 10.21%). At the same time, an increase in porosity from 3–5% to 10–12% led to a deterioration in the quality of the appearance of the coatings.

Putting the surface of the substrate with a porosity of 6%, followed by priming, increased the quality class of the appearance of the coatings (Table 6).

The quality of the appearance of coatings was also determined by the method of applying the paint composition. Table 6 provides information about the quality of the coatings' appearances, which depended on the method of applying the paint composition for preparing the surface of the substrate. An analysis of the data given in Table 6 shows that, with an equal porosity of the cement substrate, the pneumatic application of the paint composition contributed to an increase in the coating class. Thus, in order to obtain a higher quality in the appearance of the coating, it is necessary to strive to create a surface porosity of the cement substrate that does not exceed 5%.

**Table 5.** Effect of substrate porosity on the appearance quality of coatings.

| Coating Type | Porosity, % | Coating Roughness, μm/Coefficient of Variation, % | Coating Grade |
|---|---|---|---|
| | 0 | 33.9/4.9 | V |
| Lime | 5.7 | 50.5/5.1 | V |
| | 9.6 | 83.4/5.2 | VI |
| | 0 | 33.5/4.4 | V |
| Polymer lime | 4.1 | 41.3/4.5 | V |
| | 11.9 | 88.1/4.9 | VI |
| | 0 | 8.2/4.1 | V |
| PVAC | 3.1 | 20.5/4.3 | IV |
| | 12.6 | 58.4/4.7 | V |

**Table 6.** Influence of the substrate surface preparation method on quality appearance of coatings.

| Viscosity Determined by VZ-4, s | Paint Application Method | Porosity, % | Substrate Surface Preparation | Coating Appearance Quality Grade According to Russian Standard GOST 9.032-74 |
|---|---|---|---|---|
| | | Polyvinyl Acetate Cement Paint | | |
| 50 | brush | 0 | padding | VI |
| 30 | pneumatic | 0 | padding | V |
| 50 | brush | 3.2 | padding | VI |
| 50 | brush | 4.3 | padding | VI |
| 50 | brush | 6 | padding | V |
| 50 | brush | 6 | puttying, padding | IV |
| | | Lime paint | | |
| 50 | brush | 0 | padding | VI |
| 35 | pneumatic | 0 | padding | V |
| 50 | brush | 6 | padding | VI |

## 4. Conclusions

An assessment of the quality of the appearance of some composite coatings using fractal dimension has been carried out. The following conclusions were made, emphasizing scientific originality and practical relevance:

1.  A correlation was established between the surface roughness of the coating, the quality grade of their appearance, and the fractal dimension.
2.  A model of the coating surface profile length with fractal dimension $D$ is proposed. It is suggested to evaluate the quality of the surface of paint and varnish coatings by the fractal dimension index.
3.  The relationship between the numerical values of the fractal index and the quality grade of the appearance of coatings was established.
4.  The patterns of formation in the quality of the appearance of coatings from the rheological and technological properties of paint compositions were established. It was revealed that with an increase in the surface tension of the aqueous paint composition, a lower quality of the appearance of the resulting coating was observed. With an increase in the porosity of the substrate, an increase in the roughness of the surface of the coatings was observed.
5.  Numerical values of the index of the fractal dimension of the coating surface profile were obtained and depended on the porosity of the substrate.
6.  The results obtained can be applied in various types of production to improve the quality of paint coatings.

**Author Contributions:** Conceptualization, V.L. (Valentina Loganina), V.L. (Valery Lesovik) and R.S.M.R.; Data curation, R.F., V.O., I.M., D.N.Q., M.L. and O.L.; Formal analysis, V.L. (Valentina Loganina); Investigation, R.F.; Resources, M.A.; Software, V.O., I.M., D.N.Q., M.L. and O.L.; Writing—original draft, R.S.M.R., V.L. (Valentina Loganina), R.F., V.L. (Valentina Loganina), V.L. (Valery Lesovik) and I.M. All authors have read and agreed to the published version of the manuscript.

**Funding:** The study was supported by the RSF grant No. 22-19-20115, https://rscf.ru/project/22-19-20115/ (accessed on 18 November 2022) and the Government of the Belgorod Region, Agreement No. 3 of 24 March 2022.

**Institutional Review Board Statement:** Not applicable.

**Informed Consent Statement:** The research did not involve Human Participants and/or Animals.

**Conflicts of Interest:** The authors declare no conflict of interest.

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
