# Peer review of "Evaluation of Some Composite Paint Coatings’ Appearance Quality Using Fractal Dimension"

_jcs, doi:10.3390/jcs7010009_

Round 1

Reviewer 1 Report (Previous Reviewer 2)

The authors have significantly improved the paper in this re-submitted version.

The introduction can be improved by relating to the mechanics of the studied materials and their mechanical characteristics. The reference to be included is: 10.1177/00219983221141154.

Author Response

Dear Reviewer 1!

Thank you for your interest in our manuscript. Your valuable comments helped make our manuscript even better. All corrections in the manuscript are highlighted in blue.

Comment 1:  The introduction can be improved by relating to the mechanics of the studied materials and their mechanical characteristics. The reference to be included is: 10.1177/00219983221141154.

Response: This information has been described in the introduction using the article kindly recommended to you.

Reviewer 2 Report (New Reviewer)

1. The article studies the appearance quality of the coating. For composite coatings in different situations, some pictures should be used to increase the readability of the article.

2.The clarity of Figure 1 is too poor, please use a higher resolution picture instead.

3. The formula format should be consistent (eg. the font size of formula (3) and formula (5)). Likewise, the relevant formatting of the figures and figures in the text should be improved.

4. The main conclusions obtained should be further proved by verification experiments or multi-objective optimization experiments.

Author Response

Dear Reviewer 2!

Thank you for your interest in our manuscript. Your valuable comments helped make our manuscript even better. All corrections in the manuscript are highlighted in blue.

Comment 1: The article studies the appearance quality of the coating. For composite coatings in different situations, some pictures should be used to increase the readability of the article.

Response: The quality of the surface of the paintwork can be assessed, including its roughness, i.e. surface profile. In our opinion, the methods of fractal physics can be applied to evaluate the coating surface profile. The successful application of fractal analysis methods in the study of polymers allows us to hope that this approach will be successfully extended to assess the quality of the appearance of coatings. Meanwhile, the analysis of scientific and technical literature shows that these issues remain unresolved. In this regard, the development of an integral indicator of the quality of the surface of the paintwork is of considerable scientific and practical interest. The use of an integral indicator of the quality of the appearance of coatings (fractal dimension) will allow, in addition to existing indicators, to conduct a more objective assessment of the quality of paint and varnish coatings.

Comment 2: The clarity of Figure 1 is too poor, please use a higher resolution picture instead.

Response: This figure has been replaced with a better resolution one

Comment 3: The formula format should be consistent (eg. the font size of formula (3) and formula (5)). Likewise, the relevant formatting of the figures and figures in the text should be improved.

Response: The formula format has been consistent. Likewise, the relevant formatting of the figures in the text have been improved.

Comment 4: The main conclusions obtained should be further proved by verification experiments or multi-objective optimization experiments.

Response: In this work, we set a goal, in addition to the existing indicators for assessing the quality of the appearance of coatings, to use an indicator of fractal dimension, which would make it possible to more fully assess the quality of the paintwork.

Round 2

Reviewer 1 Report (Previous Reviewer 2)

The paper can be accepted. 

Reviewer 2 Report (New Reviewer)

The physical picture of the actual situation should be added to increase the readability of the article

This manuscript is a resubmission of an earlier submission. The following is a list of the peer review reports and author responses from that submission.

Round 1

Reviewer 1 Report

This is the review report for the resubmitted manuscript no. jcs-1877202. The authors provided answers to the comments from the previous submissions, and for renewed questions the authors just copy-pasted the answers and do not respond to the comments in a substantive manner. The changes in the manuscript are again minor and do not correspond with the given suggestions. In the response to comment 1, the authors wrote that “The introduction was completely rewritten…”, while only one paragraph was added and the rest remained unchanged. In the case of other comments the authors disagree with my opinions, which, of course has sense, when a constructive feedback is given or answered in the same way as previously, without making suggested comments or explaining substantively the reasons of not considering them.

In these circumstances I uphold my decision, since the manuscript does not present novelty (well-known methods applied for the investigated materials), is written with a low quality, has no justification of generality of the proposed approach, and no statistical analysis of the results, which, in my opinion, are absolutely necessary for justifying the results.

Reviewer 2 Report

The paper presents an interesting approach based on the Evaluation of some composite paint coatings appearance quality using the fractal dimension. However, the innovation of the current research work should be further highlighted and emphasized. At the same time, the authors should consider the following comments to greatly improve the quality of the paper.

1. The title is too long. Kindly consider shortening it a little bit.

2. In the abstract, instead of the generic start "Composite materials are characterized by multiple layers, which leads to complexity in the design to ensure the effective operation of the constituent elements.", Kindly add an introductory statement that defines the existing problem.

3. The introduction needs to be improved by relating to the mechanics of the studied materials and their mechanical characteristics. The references to be included are: 10.1016/j.jiec.2022.06.023, 10.1016/j.polymertesting.2017.09.009, 10.1016/j.compstruct.2021.114698, 10.1002/app.46770, 10.3390/polym14132662, 10.1016/j.porgcoat.2022.107015.

4. In Section 2.1. Characteristics of the Raw Materials Used, kindly add a table that shows the main chemical and physical properties of the raw materials used in the present study.

5. In the laboratory equipment section, the surface roughness measurement procedure hasn't been described. The machine used has to be mentioned, the standard followed has to be declared and the roughness parameters assessed has to be described and justified.

6. It is recommended to present a conclusive part that defines the suitability of each coating type for given applications.

7. The conclusion needs to be modified to summarize the research outcomes in short statements with clear observations.